# Resilience and Mobbing Among Nurses in Emergency Departments: A Cross-Sectional Study

**DOI:** 10.3390/healthcare13151908

**Published:** 2025-08-05

**Authors:** Aristotelis Koinis, Ioanna V. Papathanasiou, Ioannis Moisoglou, Ioannis Kouroutzis, Vasileios Tzenetidis, Dimitra Anagnostopoulou, Pavlos Sarafis, Maria Malliarou

**Affiliations:** 1Laboratory of Education and Research of Trauma Care and Patient Safety, Department of Nursing, University of Thessaly, Gaiopolis Campus, Larissa-Trikala Ring-Road, 415 00 Larissa, Greece; ikouroutzis@uth.gr (I.K.); vasileiostzen@gmail.com (V.T.); danagnostop@uth.gr (D.A.); psarafis@uth.gr (P.S.); malliarou@uth.gr (M.M.); 2Department of Nursing, University of Thessaly, Gaiopolis Campus, Larissa-Trikala Ring-Road, 415 00 Larissa, Greece; iomoysoglou@uth.gr

**Keywords:** mobbing, mental resilience, nurses, hospital, emergency department, work environment, prevention and treatment of ethical harassment

## Abstract

Background: Moral harassment (mobbing) in healthcare, particularly among nurses, remains a persistent issue with detrimental effects on mental health, resilience, and quality of life. Aim: We examine the relationship between the resilience of nurses working in Emergency Departments (EDs) and how these factors influence experiences of workplace mobbing. Methods: This cross-sectional study included 90 nurses from four public hospitals in Greece’s 5th Health District. Data were collected between October 2023 and March 2024 using the WHOQOL-BREF, Workplace Psychologically Violent Behaviors (WPVB) scale and the Connor–Davidson Resilience Scale (CD-RISC). The sample consisted primarily of full-time nurses (84.3% female; mean age = 43.1 years), with 21.1% reporting chronic conditions. Most participants were married (80.0%) and had children (74.4%), typically two (56.1%). Statistical analyses—conducted using SPSS version 27.0—included descriptive statistics, Pearson and Spearman correlations, multiple linear regression, and mediation analysis, with significance set at *p* < 0.05. Results: Resilience was moderate (mean = 66.38%; Cronbach’s α = 0.93) and positively correlated with all WHOQOL-BREF domains—physical, psychological, social, and environmental (r = 0.30–0.40)—but not with the overall WHOQOL-BREF. The mean overall WHOQOL-BREF score was 68.4%, with the lowest scores observed in the environmental domain (mean = 53.76%). Workplace mobbing levels were low to moderate (mean WPVB score = 17.87), with subscale reliabilities ranging from α = 0.78 to 0.95. Mobbing was negatively associated with social relationships and the environmental WHOQOL-BREF (ρ = –0.23 to –0.33). Regression analysis showed that cohabitation and higher resilience significantly predicted better WHOQOL-BREF outcomes, whereas mobbing was not a significant predictor. Mediation analysis (bootstrap N = 5000) indicated no significant indirect effect of resilience in the relationship between mobbing and WHOQOL-BREF. Conclusions: Resilience was identified as a key protective factor for nurses’ quality of life in emergency care settings. Although workplace mobbing was present at low-to-moderate levels, it was negatively associated with specific WHOQOL-BREF domains. Enhancing mental resilience among nurses may serve as a valuable strategy to mitigate the psychological effects of moral harassment in healthcare environments.

## 1. Introduction

Nursing professionals in Emergency Departments (EDs) work under immense pressure due to high patient turnover, tight time constraints, and substantial emotional demands. These challenging conditions often foster workplace mobbing—a pattern of repeated, hostile behaviors intended to undermine an individual’s dignity and psychological well-being [1].

Initially coined by Leymann in 1984, the term mobbing refers to systematic, unethical, and aggressive conduct by one or more individuals aimed at a colleague [2,3]. “Over the past decades, workplace mobbing has gained recognition as a serious social issue with significant implications for both physical and mental health.” Over the past decades, it has gained recognition as a serious social issue with significant implications for both physical and mental health. Research consistently shows that high-pressure environments—particularly in healthcare—are especially vulnerable to such behaviors [2,3,4,5,6,7,8]. Consequences can include emotional distress, psychosomatic symptoms, burnout, and diminished work performance.

In Greece, although mobbing is not institutionally recognized, labor unions such as the General Confederation of Greek Workers (GSEE) have advocated for its inclusion in collective labor agreements [8]. Socioeconomic instability in recent years has complicated the issue; job security concerns have made workers more tolerant of abusive behaviors. Nevertheless, safeguarding human dignity remains a key concern for researchers and labor advocates seeking to improve workplace conditions [9,10].

### 1.1. Literature Review

In Greece, high rates of mobbing have been documented in the healthcare sector [11]. A study across seven hospitals under the 6th Health Insurance Fund reported that 71% of nurses had experienced moral harassment in the past year. A large proportion of these nurses suffered from psychosomatic symptoms such as anxiety, headaches, depression, and negative attitudes toward their work [11]. Furthermore, 80% of those affected indicated that their personal lives were negatively impacted, yet only 38.2% sought professional help.

An earlier 2004 survey of the ED found that 85.8% of healthcare workers had experienced some form of harassment within the preceding year [12,13]. A significant number of incidents were caused by patient relatives due to delays in care delivery, while 24% stemmed from interpersonal conflicts among colleagues. Notably, 67% of staff managed such conflicts independently, with minimal managerial support, which contributed to feelings of insecurity and dissatisfaction [14].

A comparative study of public and private hospitals revealed similarly high levels of mobbing in both settings, underscoring the pervasive nature of the issue across the healthcare spectrum [15].

The international literature reflects similar findings across a wide range of countries. Studies conducted in Europe (Sweden, France, Italy, and Turkey) and beyond (USA, Australia, and China) highlight the widespread prevalence of moral harassment in various professional environments, including law enforcement and private-sector workplaces, in addition to healthcare [2,16,17,18,19,20,21,22].

In Turkey, Yildirim and Yildirim [2] found that 86.5% of nurses in both public and private hospitals had experienced workplace harassment in the past year. Nurses in private hospitals were more frequently targeted, primarily by their supervisors. Similarly, Sahin et al. [16] studied junior male physicians and reported that 87.7% had experienced mobbing, with higher incidence among those working over 40 h per week, those who were single, and those employed in university or private hospitals. Efe and Ayaz [17] also reported frequent mobbing, with nursing service directors often identified as perpetrators and communication breakdowns cited as a major contributing factor.

Çevik Akyil et al. [18] examined nurses in Eastern Turkey, finding that younger, less experienced nurses and those working night shifts were particularly vulnerable. Managers were again frequently identified as the instigators, with poor working conditions acting as a catalyst. In Taiwan, Pai and Lee [19] found that 51.4% of nurses reported verbal abuse, 19.6% had experienced physical violence, and 12.9% had been subjected to sexual harassment. Risk factors included being under the age of 30 and working night shifts.

In North America, the ED is considered among the most hazardous healthcare environments. A U.S. study revealed that the ED was the most common setting for physical assaults and the second most common for homicides involving healthcare professionals. Between 1980 and 1990, 35 healthcare workers lost their lives on duty [20]. Presley et al. [21], in a study involving 1209 ED nurses in Pennsylvania, reported that 97% had encountered some form of harassment during their careers.

In Australia, Lyneham [22] surveyed 266 ED nurses in New South Wales, all of whom reported experiencing workplace violence at least weekly. Notably, 92 incidents involved lethal weapons or threats, with 92% of perpetrators identified as patients or their companions.

### 1.2. Mobbing, Mental Resilience, and WHOQOL-BREF Among Nursing Staff in Emergency Departments

ED nurses are especially vulnerable to mobbing due to hierarchical pressures, chronic understaffing, and poor organizational climate. Consequences include emotional exhaustion, anxiety, depression, and reduced job satisfaction [23,24]. Havaei et al. [25] linked workplace violence directly to poor mental health, while Longo [26] emphasized the role of ineffective leadership in enabling mobbing.

### 1.3. Mental Resilience as a Protective Factor

Mental resilience—the capacity to adapt to stress and recover from adversity—has been identified as a vital buffer against the psychological harm caused by mobbing [27]. “Resilient nurses generally experience lower levels of emotional distress, apply more effective coping strategies, and maintain higher professional performance despite prolonged exposure to workplace stressors [28,29].” Mealer et al. [30], focusing on ICU and ED nurses, demonstrated that resilience training significantly reduced PTSD symptoms and improved coping with hostile work environments.

Siriwardena et al. [31] also found a strong positive correlation between resilience and the ability to maintain professional efficacy and psychological well-being in high-stress ED settings.

### 1.4. WHOQOL-BREF Implications

Workplace harassment negatively affects nurses’ overall WHOQOL-BREF, especially mental health and interpersonal relationships [27,32]. However, resilience may mitigate these outcomes. Zarei et al. [33] found that resilience-enhancing interventions improved psychological well-being and job satisfaction among nurses exposed to mobbing.

### 1.5. Significance and Importance of the Study

Emergency Departments are among the most demanding and high-stakes environments in the healthcare system. They are not only sites for urgent clinical interventions but also complex psychosocial settings where staff are routinely exposed to trauma, unpredictability, and intense emotional labor. Nurses in the ED are particularly vulnerable, often facing cumulative burdens of physical fatigue, emotional strain, and psychological stress due to the high-intensity nature of their work [32,33]. Shift work, understaffing, time pressure, and repeated exposure to critical incidents contribute to chronic fatigue and burnout [34,35]. These stressors are further exacerbated by frequent incidents of workplace violence—from patients, relatives, or colleagues. Unfortunately, such incidents are often underreported due to fear of retaliation, institutional inertia, or diminished professional self-efficacy [36,37]. The consequences are serious and include emotional exhaustion, job dissatisfaction, and increased intentions to leave the profession [38,39]. Additionally, organizational culture and psychological safety play a critical role in how such challenges are internalized and managed by staff [40].

Within this volatile environment, mobbing—or systematic moral harassment—emerges as a particularly destructive form of workplace aggression. It not only affects the mental and physical health of individuals but also weakens team dynamics and undermines healthcare delivery [39,41]. Despite its severity, mobbing remains under-investigated in Greece, particularly among frontline staff such as ED nurses, with minimal institutional mechanisms to prevent or mitigate it [24,41].

This study addresses this research gap by highlighting resilience as a central protective factor in managing the negative psychosocial outcomes of mobbing. Importantly, resilience is not presented merely as a static trait but as a modifiable capacity that can be enhanced through targeted interventions [29,42].

Based on our findings, we propose the development of psychoeducational and workplace-based programs focused on the following:Strengthening psychological resilience;Fostering a greater sense of purpose and professional identity;Improving overall WHOQOL-BREF scores [43,44].

These interventions may include structured group workshops, stress management training, peer support systems, and reflective clinical practice, all grounded in evidence-based approaches from counseling psychology and occupational health [45,46,47,48].

### 1.6. Scope and Aims of the Research

This study explores the relationship between psychological resilience and WHOQOL-BREF in the context of moral harassment (mobbing) experienced by nurses working in the Emergency Department (ED) of public hospitals within Greece’s 5th Regional Health Authority. The primary aim is to examine the potential protective role of resilience in mitigating the negative consequences of moral harassment, with the broader goal of identifying strategies to support nurses’ psychological well-being and professional sustainability.

More specifically, the research seeks to carry out the following:

Assess the prevalence and nature of mobbing incidents within the ED;

Evaluate the psychological and emotional effects of mobbing on nursing personnel;

Investigate how resilience moderates the impact of mobbing on nurses’ perceived WHOQOL-BREF;

Provide evidence-based recommendations for interventions that strengthen psychological resilience and improve overall WHOQOL-BREF among ED nurses.

## 2. Materials and Methods

### 2.1. Study Design

This study employed a cross-sectional, non-experimental, and descriptive design, appropriate for investigating relationships between psychological and workplace variables at a specific point in time. Data were collected once per participant over a defined period (October 2023 to March 2024), reflecting a snapshot of conditions experienced by Emergency Department (ED) nurses across four public hospitals.

A synchronic (single time-point) approach was used to collect data on three key variables—workplace mobbing, psychological resilience, and quality of life—simultaneously. This design enables the assessment of prevalence, associations, and group differences without implying causation [34]. The absence of intervention or follow-up categorizes this study firmly within a cross-sectional observational framework.

This methodological choice was made due to the practical constraints of the high-intensity clinical environment and the need for a feasible, minimally disruptive research strategy. The descriptive component of the design aimed to document the frequency and characteristics of workplace mobbing, while the analytical aspect explored how resilience and quality of life were related to mobbing experiences.

Although cross-sectional designs do not allow for causal inferences, they are effective for identifying patterns and correlations that can inform future longitudinal or interventional research aimed at preventing or mitigating moral harassment in healthcare settings.

This design was appropriate given the resource constraints and the dynamic, high-pressure nature of emergency healthcare settings. While it does not infer causality, it allows for the identification of potential correlations between variables [34,35,40]. The descriptive nature of the study aimed to document the prevalence and characteristics of workplace mobbing and explore related psychosocial outcomes. The findings are intended to inform future interventions aimed at preventing or addressing moral harassment in similar healthcare contexts.

### 2.2. Place of Conduct

The research was conducted in the Emergency Departments of four public hospitals under the jurisdiction of the 5th Regional Health Authority (DYPE), which covers Thessaly and Central Greece.

Hospitals were selected based on the following explicit inclusion criteria:High volume of emergency admissions, ensuring exposure to high-stress clinical environments.Presence of full-time ED nursing staff, ensuring consistency in participants’ work settings.Institutional willingness to participate, including administrative approval and ethical clearance.Variability in organizational structure and culture, allowing for a broader representation of workplace conditions.

Although all hospitals function under the same regional authority, they differ in staffing models, institutional culture, and patient demographics—all factors that may influence the nature and perception of workplace mobbing. Including multiple institutions enhanced the ecological validity and generalizability of the study.

### 2.3. Study Population

The study included 90 full-time nursing professionals (registered nurses and nurse assistants) working in the Emergency Departments of the selected hospitals. A convenience sampling method was used, due to logistical constraints and the unpredictable workflow in emergency settings [37]. Despite this limitation, the sample represents a diverse cross-section of professionals involved in emergency care across multiple institutions.

### 2.4. Study Instruments

Three questionnaires were used in this study to assess mobbing, quality of life, and resilience among healthcare professionals.

#### 2.4.1. Workplace Psychological Violence Behavior (WPVB) Questionnaire

The Workplace Psychologically Violent Behaviors (WPVB) instrument, originally developed by Yildirim and Yildirim [39], was employed to assess the frequency and nature of psychological violence experienced by healthcare professionals. This comprehensive self-report tool consists of 33 items, structured across four core domains that reflect different dimensions of psychological violence in the workplace:

Isolation from work (11 items): This subscale captures behaviors that distance the individual from work-related communication, participation, and inclusion. Attacks on professional role and performance (9 items): This dimension measures verbal or symbolic assaults targeting the worker’s professional competence and role. Aggressive behaviors toward the person (9 items): This includes personal hostility or threats directed at the individual rather than their professional identity. Direct negative acts (4 items): This subscale addresses overt, harmful actions such as shouting or public humiliation. Participants rated each item on a 6-point Likert scale ranging from 0 (“never”) to 5 (“always”), indicating the frequency with which they experienced each behavior in the workplace over a defined period.

The WPVB scale has demonstrated excellent internal consistency, with an overall Cronbach’s alpha (α) coefficient of 0.93. The subscales also exhibit strong reliability, with alpha values ranging from 0.70 to 0.91, affirming the psychometric robustness of the tool across its domains. For the purposes of this study, the instrument was culturally adapted and linguistically validated for use in the Greek context, following standard procedures for translation and back-translation. Formal permission for both the use and adaptation of the tool was obtained from the original authors and relevant institutional bodies [41].

#### 2.4.2. WHOQOL-BREF

To evaluate participants’ perceived quality of life, the WHOQOL-BREF instrument was employed. Developed by the World Health Organization (WHO) [43], this standardized tool is a widely recognized and validated measure of multidimensional well-being. It consists of 30 items, encompassing four primary domains:

Physical health and independence: measures energy levels, mobility, daily activities, sleep, and work capacity. Psychological health and spirituality: assesses body image, self-esteem, negative and positive feelings, and personal beliefs. Social relationships: evaluates personal relationships, social support, and sexual activity. Environmental factors: includes questions on financial resources, health care access, home environment, safety, leisure, and transportation.

Each domain is scored on a scale from 4 to 20, with higher scores indicating a better perceived quality of life in that domain. The WHOQOL-BREF is suitable for use across cultures and has been validated in numerous international settings.

For this study, the Greek version of the WHOQOL-BREF, validated by Tzinieri-Kokkosi et al. [44], was utilized. This version has been psychometrically tested for use with Greek-speaking populations and retains strong reliability and validity indices. Official authorization to use this instrument in the study was obtained.

#### 2.4.3. Connor–Davidson Resilience Scale (CD-RISC)

The Connor–Davidson Resilience Scale (CD-RISC-25) is a widely used, psychometrically validated instrument designed to measure psychological resilience, defined as the capacity to cope effectively with adversity, stress, and traumatic events. Originally developed by Connor and Davidson [42], the full version of the scale contains 25 self-report items, each rated on a 5-point Likert scale, ranging from 0 to 4. The total score ranges from 0 to 100, with higher scores indicating greater levels of resilience.

The CD-RISC-25 assesses five core dimensions of resilience:

Personal competence, high standards, and tenacity—reflecting self-efficacy, persistence, and confidence in one’s problem-solving abilities.

Trust in one’s instincts and tolerance of negative affect—encompassing adaptive emotional processing and intuitive decision-making.

Positive acceptance of change and secure relationships—indicating flexibility, adaptability, and the ability to maintain meaningful interpersonal connections.

Sense of control—measuring one’s perceived influence over life circumstances.

Spiritual influences—capturing the role of faith or existential beliefs as sources of strength.

The instrument demonstrates excellent internal consistency, with a Cronbach’s alpha (α) of 0.89, and strong test–retest reliability, with an intraclass correlation coefficient (ICC) of 0.87, confirming its stability over time.

For this study, the Greek translation of the CD-RISC-25 was used.

### 2.5. Data Collection Process-Research Ethics

Ethical approval was obtained from the Scientific Councils of participating hospitals. The study was explained to department heads, and questionnaires were distributed in person by the primary researcher. Participation was voluntary, anonymous, and confidential. Completion time was 15–30 min, and questionnaires were returned within 2–4 weeks. All ethical principles, including those outlined in the Declaration of Helsinki (2004 amendment) [49], were upheld.

### 2.6. Statistical Analysis

The normality of the quantitative variables’ distribution was assessed via the Kolmogorov–Smirnov criterion. The assumption of normality was not found to be valid across all WPVB scales. Quantitative variables were expressed as mean values (standard deviation) and as medians (interquartile range), while categorical and ordinal variables were expressed as absolute and relative frequencies.

The association between WHOQOL-BREF and CD-RISC (resilience) scales was evaluated via Pearson correlation coefficient (r), while the relationship between WHOQOL-BREF and WPVB scales was examined using Spearman’s rank correlation coefficient (rho) due to non-normal distributions. Correlation coefficients were interpreted as follows: very high (>0.9), high (0.7–0.9), moderate (0.5–0.7), low (0.3–0.5), and very low (<0.3) [48].

Multiple linear regression analysis was used to examine predictors of the WHOQOL-BREF domain scores. Adjusted regression coefficients (β) with standard errors (SE) were reported for each model.

To investigate the mediating role of psychological resilience in the relationship between workplace psychological violence and quality of life, the PROCESS macro for SPSS (version 4.2) was used, in accordance with Hayes’ guidelines [50]. A 5000-sample bootstrap procedure was applied to estimate bias-corrected 95% confidence intervals (CI) for the indirect effects. Mediation was considered statistically significant when the confidence intervals did not include zero. Full mediation was indicated when the direct effect was non-significant; partial mediation was present when both direct and indirect effects were significant [51,52,53,54,55].

Internal consistency reliability was assessed using Cronbach’s alpha coefficient for each questionnaire and subscale. A threshold of α ≥ 0.70 was considered acceptable.

Scoring for each questionnaire followed standardized procedures.

The WPVB subscale scores were computed by summing the responses within each domain (range per item: 0–5), resulting in both individual domain and overall scores.The CD-RISC-25 total resilience score was calculated by summing the 25 items (range: 0–100), with higher values reflecting greater resilience.The WHOQOL-BREF domain scores were computed according to the WHO scoring manual and were transformed into a 0–100 scale, with higher scores indicating a better perceived quality of life.

Data completeness was verified prior to statistical analysis. All participants completed the entire set of questionnaires, including demographic information, with no missing values. This ensured the robustness and reliability of the analyses, without the need for imputation or data exclusion.

All reported *p*-values were two-tailed, and statistical significance was set at *p* < 0.05. Analyses were performed using SPSS statistical software (version 27.0).

These methodological and analytical strategies ensured the validity, reliability, and interpretability of findings, while minimizing bias and maximizing the study’s applicability in real-world healthcare settings.

To facilitate the interpretation of hypothesized relationships, a conceptual model was developed and is presented in Figure 1. In this model, workplace psychological violence (WPVB) is posited as the independent variable, resilience (CD-RISC) as the mediating variable, and quality of life (WHOQOL-BREF) as the outcome variable. The model was tested using mediation analysis based on Hayes’ PROCESS macro.

## 3. Results

A total of 120 self-administered questionnaires were distributed to eligible participants, of which 90 were completed and returned, yielding a response rate of 75%. The sample size was determined in accordance with Cohen’s statistical power guidelines for social sciences, with adjustments made for anticipated non-response to ensure sufficient statistical validity and representativeness.

The majority were female (84.3%) with a mean age of 43.1 years (SD = 8.4). Most participants held a technological university degree (71.1%), while smaller percentages held MSc degrees (13.3%), two-year college diplomas (6.7%), university degrees (5.6%), or PhDs (3.3%). In terms of family status, 80% were married, 17.8% unmarried, and 2.2% divorced. A substantial proportion (74.4%) had children, with the majority (56.1%) having two children, 34.8% having one, and 9.1% having three children. The vast majority lived with others (84.4%). Professionally, nearly all (98.9%) were registered nurses. About 21.1% reported having chronic health issues, including arthritis, heart problems, hypertension, cancer, and diabetes. The most commonly reported conditions included arthritis or rheumatism (31.6%), followed by heart problems (15.8%), hypertension (15.8%), and cancer (15.8%). Additional conditions included leg problems (10.5%), diabetes (5.3%), and other unspecified ailments (10.5%) (Table 1).

The mean “physical health” score was 65.56% (SD = 12.53%) and mean “psychological health” was 64.31% (SD = 12.22%), Table 2. The mean “social relationships” score was 66.56% (SD = 13.25%) and the mean “environment” score was 53.76% (SD = 12.99%). Acceptable reliability was present in all under study scales, since the alpha coefficients were over 0.70. The mean “resilience” score was 66.38 (SD = 12.83%). The mean overall quality of life (QoL) score, as assessed by the WHOQOL-BREF, was 68.4% (SD = 14.49), indicating a moderate to high level of perceived well-being among participants. This overall score reflects a composite of multiple WHOQOL-BREF dimensions including physical health, psychological state, social relationships, and environmental factors.

As far as the WPVB scale is concerned, it was found that a greater score in the factor “individual’s isolation from work” was significantly associated with worse social relationships (rho = −0.23; *p* = 0.030) and a worse environment (rho = −0.33; *p* = 0.002), Table 3. Also, a greater score in the factor “direct attack” was significantly correlated with a worse environment (rho = −0.30; *p* = 0.005) and so was the total WPVB score (rho = −0.28; *p* = 0.008). The resilience score was significantly and positively correlated with all WHOQOL-BREF subscales (r coefficients ranged from 0.30 to 0.40), except for the overall WHOQOL-BREF (r = 0.14; *p* = 0.189).

Also, greater resilience was significantly associated with better physical (β = 0.25; *p* = 0.023) and psychological health (β = 0.24; *p* = 0.006), Table 4. The total WPVB score as well as its factors were not significantly correlated with the overall WHOQOL-BREF, physical health, and psychological health scores (*p* > 0.05); thus, there is no need for further analysis regarding the mediating role of resilience in the association between mobbing and WHOQOL-BREF. Participants living with others had significantly greater overall WHOQOL-BREF (β = 13.1; *p* = 0.019), physical health (β = 12.41; *p* = 0.011), and psychological health scores (β = 13.51; *p* = 0.001).

Only the resilience score was found to be significantly associated with social relationships and the environment, as presented in Table 5. More specifically, greater resilience was significantly associated with better social relationships (β = 0.32; *p* = 0.005) and a better environment (β = 0.27; *p* = 0.024). When WPVB subscales were entered into the analyses, in turn, instead of the total score, no significant results were found (*p* > 0.05).

The resilience score was not found to significantly mediate the correlation between the total WPVB score and “environment” [indirect effect (95% CI) = −0.053 (−0.153, 0.010) and direct effect (95% CI) = −0.012 (−0.162, 0.138)], as presented in Table 6. Similarly, the resilience score was not found to significantly mediate the correlation between the “direct attack” score and “environment” [indirect effect (95% CI) = −0.143 (−0.740, 0.172) and direct effect (95% CI) = −0.359 (−0.740, 0.172)], nor the correlation between the “individual’s isolation from work” score and “environment” [indirect effect (95% CI) = −0.124 (−0.371, 0.043) and direct effect (95% CI) = −0.153 (−0.550, 0.245)]. Also, the resilience score was not found to significantly mediate the correlation between the “individual’s isolation from work” score and “social relationships” [indirect effect (95% CI) = −0.148 (−0.464, 0.026) and direct effect (95% CI) = 0.057 (−0.317, 0.431)]. *p*-values are approximate and inferred from the 95% confidence intervals provided.

## 4. Discussion

The present study aimed to explore the interrelationships between workplace bullying (mobbing), psychological resilience, and WHOQOL-BREF among nursing professionals working in Emergency Departments. A key objective was to assess whether resilience functions as a mediating variable, potentially buffering the negative impact of mobbing on nurses’ perceived quality of life.

The findings of this study revealed several important patterns that enhance our understanding of the complex psychosocial dynamics affecting nurses in high-stress clinical environments. Specifically, the data suggest that exposure to mobbing behaviors is significantly associated with lower WHOQOL-BREF scores, particularly in the domains of psychological well-being and social relationships. Conversely, higher levels of resilience were positively correlated with improved WHOQOL-BREF outcomes and appeared to mitigate the detrimental effects of workplace harassment.

These results underscore the critical role of resilience as a protective psychological mechanism, suggesting that interventions aimed at fostering resilience may be effective in improving overall well-being and professional satisfaction among nurses exposed to workplace adversity. In doing so, this study contributes to the broader literature by emphasizing the importance of organizational strategies that not only reduce bullying behaviors but also strengthen individual coping resources within healthcare environments.

### 4.1. WHOQOL-BREF and Resilience

Resilience was positively and significantly associated with most WHOQOL-BREF domains, including physical and psychological health, social relationships, and environmental perception. These findings are in line with previous research demonstrating the buffering role of resilience in occupational stress and its contribution to improved well-being and functioning among nurses and healthcare professionals [28,29]. Specifically, our data revealed moderate correlations between resilience and WHOQOL-BREF domains, particularly psychological health (r = 0.40) and social relationships (r = 0.35), suggesting that resilience may help to mitigate the psychological toll of challenging work environments. In a study of 280 hospital nurses in Istanbul, psychological resilience was positively correlated with compassion satisfaction (r = 0.372) and negatively correlated with burnout (r = −0.379) and compassion fatigue (r = −0.336), indicating that resilience contributes to a higher professional WHOQOL-BREF [56].

However, the lack of a significant correlation between resilience and overall WHOQOL-BREF (r = 0.14, *p* = 0.189) contrasts with prior studies where resilience was consistently found to be a strong predictor of global life satisfaction [57,58]. This discrepancy may be attributed to contextual or cultural factors, sample-specific characteristics, or the potential for resilience to exert a more domain-specific rather than global effect on WHOQOL-BREF.

### 4.2. Mobbing and WHOQOL-BREF

Our findings demonstrated that certain dimensions of workplace bullying—namely, “individual’s isolation from work” and “direct attack”—were associated with reduced scores in specific WHOQOL-BREF domains such as the environment and social relationships. These results are consistent with the existing literature indicating that mobbing can negatively affect nurses’ perception of their social and occupational environments [59,60]. A Greek NHS study highlighted that employees subjected to mobbing reported significantly lower WHOQOL-BREF scores, emphasizing the detrimental impact of workplace harassment on overall well-being. Notably, the “environment” domain, which encompasses physical safety, financial resources, and access to services, appeared particularly sensitive to bullying-related stressors [61].

Nonetheless, we did not observe significant associations between total WPVB scores or its subscales and the broader WHOQOL-BREF domains of physical and psychological health in our regression models. This finding diverges from prior studies where workplace bullying was strongly linked to anxiety, depression, and burnout [62,63]. One potential explanation could be the relatively low mean scores of mobbing in our sample, indicating limited exposure or underreporting of workplace bullying incidents. Alternatively, the supportive social networks reported by a majority of participants (84.4% living with others) might have provided protective effects against the psychological sequelae of bullying.

### 4.3. Resilience and Mobbing

Our findings indicated no statistically significant correlation between resilience and the overall WPVB score (r = –0.12, *p* > 0.05), nor between resilience and any of its subscales (e.g., “individual’s isolation from work,” “attack on professional status,” or “direct attack”). These results diverge from prior research, which often identifies an inverse relationship between exposure to mobbing and psychological resilience among healthcare professionals [64,65,66].

One possible explanation is that the mean scores of mobbing in our sample were relatively low, suggesting limited exposure to intense or chronic bullying. As such, resilience may not have been “activated” as a psychological defense in the same way it is under more severe workplace adversity. Moreover, 84.4% of our participants reported living with others, possibly benefitting from supportive social environments that independently buffered the impact of workplace challenges—thus reducing the observable link between mobbing and resilience.

Another consideration is the qualitative nature of mobbing reported, with subtler forms such as social exclusion or indirect undermining being more prevalent than overt aggression. These experiences may require different coping resources, such as assertiveness training or organizational intervention, rather than relying solely on personal resilience.

While our results do not support a direct relationship between mobbing and resilience, they highlight the complexity of the interplay between individual and contextual protective factors. Prior studies, such as those by Yildirim [66] and Karimi et al. [67], observed significant mediation or moderation effects of resilience, but their samples were characterized by higher levels of reported bullying or involved different cultural and institutional frameworks.

Therefore, our findings do not negate the protective role of resilience but rather suggest that its effects may be context-dependent and potentially moderated by exposure intensity, coping context, and organizational support. Longitudinal research would be valuable in better capturing these dynamic relationships.

### 4.4. Mediating Role of Resilience

The findings reveal a moderate level of resilience among the participating nurses (mean = 66.38%), with excellent internal consistency (α = 0.93). Contrary to our hypothesis and the mediation framework proposed by Hayes [50], resilience did not significantly mediate the relationship between mobbing and any of the WHOQOL-BREF domains. The bootstrapped confidence intervals of indirect effects consistently included zero, suggesting no mediation effect. This result contradicts findings by Yildirim [66] and Karimi et al. [67], who reported that resilience partially mediated the association between bullying and job satisfaction or burnout among nurses. It is possible that in our sample, resilience acted as an independent predictor of WHOQOL-BREF rather than a buffer specifically targeting the impact of mobbing. Additionally, a study on adults experiencing homelessness and mental illness found that higher resilience levels were associated with better global and mental health-related WHOQOL-BREF over time, indicating resilience’s potential moderating effect in adverse conditions [68].

Another possible explanation relates to the nature of bullying in our setting. The forms of aggression reported (e.g., isolation rather than overt attacks) may elicit different coping mechanisms or demand longer-term adaptive strategies beyond individual resilience capacity. Furthermore, our cross-sectional design precludes causal inferences, which limits the interpretation of mediation effects [69].

### 4.5. Implications for Practice and Policy

From a practical standpoint, interventions should target both individual and institutional levels. Strategies to build resilience—such as mindfulness training, reflective practice, and peer support groups—have demonstrated efficacy in supporting nurses [70].

### 4.6. Implications and Strengths

This study adds to the growing body of literature that emphasizes the importance of psychosocial factors in shaping nurses’ well-being. By identifying resilience as a robust predictor of several WHOQOL-BREF domains, our results support interventions aimed at fostering resilience through training, peer support, and institutional change. Furthermore, our findings stress the need to address workplace bullying, particularly subtle forms such as social exclusion, which may erode perceptions of social and environmental well-being.

A notable strength of this study is the use of validated tools with strong internal consistency, as evidenced by Cronbach’s alpha values exceeding 0.70 for all scales. Additionally, the use of bootstrapping techniques for mediation analysis strengthens the reliability of our findings by mitigating issues of non-normality and small sample size.

### 4.7. Limitations

This study is not without limitations. The relatively small, single-center sample may restrict generalizability, and the reliance on self-report measures raises concerns about social desirability and recall biases. Moreover, the cross-sectional design limits causal interpretations. The findings reflect associations at a single point in time and do not imply causality. Future longitudinal or experimental studies would be required to explore causal pathways Future longitudinal studies with larger and more diverse samples are warranted to further explore the mechanisms linking workplace bullying, resilience, and WHOQOL-BREF in healthcare settings.

## 5. Conclusions

This study explored the complex relationships among workplace bullying (mobbing), psychological resilience, and WHOQOL-BREF in a sample of Emergency Department nurses. While the original hypothesis posited that resilience would mediate the impact of mobbing on WHOQOL-BREF, our findings did not support a statistically significant mediating effect. However, resilience emerged as a consistent and independent predictor of enhanced psychological and social well-being—two critical domains of overall WHOQOL-BREF.

Although the associations between workplace bullying and WHOQOL-BREF were evident in specific domains—such as environmental and social aspects—these were more closely linked to subtle forms of aggression like isolation rather than overt hostility. This nuanced pattern of results suggests that resilience may function as a protective factor, but not necessarily as a mechanism that alters the direct effect of mobbing on WHOQOL-BREF. Additionally, the relatively low reported levels of mobbing and the presence of strong social support among participants may have moderated the visibility of certain relationships, limiting the detection of mediation effects.

Given the cross-sectional nature of the study, causal claims cannot be made, and the findings should be interpreted within the boundaries of the study’s methodological limitations, including self-report bias, a limited sample size, and a single-center design. Therefore, while the study offers initial insights, it does not provide definitive evidence for broad policy or intervention implementation.

That said, the results do offer preliminary empirical support for the potential value of resilience-building interventions, particularly those aimed at strengthening psychological and social functioning in high-stress healthcare environments. Moreover, the association between subtle bullying behaviors and reduced WHOQOL-BREF highlights the need for more comprehensive anti-mobbing policies that address not only overt aggression but also less visible forms of workplace hostility.

## Figures and Tables

**Figure 1 healthcare-13-01908-f001:**
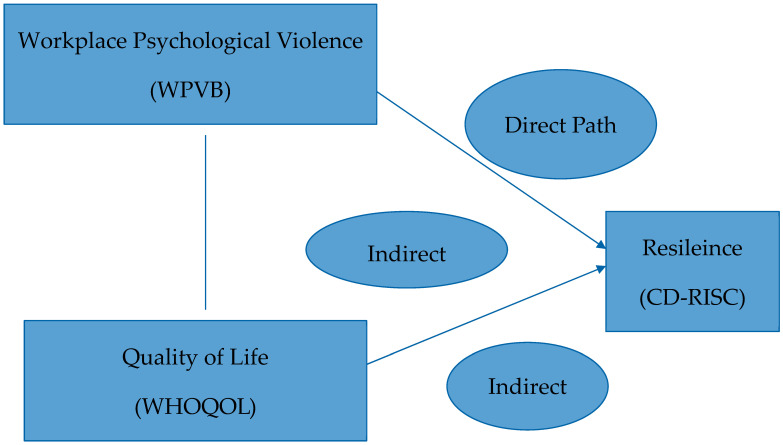
Conceptual framework of the hypothesized mediation model. Workplace psychological violence (WPVB) affects nurses’ WHOQOL-BREF potentially through the mediating effect of resilience (CD-RISC).

**Table 1 healthcare-13-01908-t001:** Sample characteristics (*n* = 90).

*n* = 90		*n*	%
Gender	Male	15	15.7
	Female	75	84.3
Educational level	2-year college	6	6.7
	Technological University (4 years)	64	71.1
	University	5	5.6
	MSc	12	13.3
	PhD	3	3.3
Family status	Single	16	17.8
	Married	72	80.0
	Divorced	2	2.2
Have children	No	23	25.6
	Yes	67	74.4
Number of children	1	23	34.8
	2	37	56.1
	3	6	9.1
Living	Alone	14	15.6
	With others	76	84.4
Professional	Nurse	89	98.9
	Assistant nurse	1	1.1
Health problems		19	21.1
If yes, define	Heart problems	3	15.8
	Arthritis or rheumatism	6	31.6
	Emphysema or chronic bronchitis	0	0.0
	Cataract	0	0.0
	Bone fracture or crack	0	0.0
	Leg problems	2	10.5
	Parkinson’s disease	0	0.0
	Hypertension	3	15.8
	Cancer	3	15.8
	Diabetes	1	5.3
	Stroke	0	0.0
	Chronic mental health problems	0	0.0
	Rectal bleeding	0	0.0
	Other	2	10.5
		Mean	SD
Age		43.1	8.4

**Table 2 healthcare-13-01908-t002:** Descriptive measures for CD-RISC, WPVB, and WHOQOL-BREF scales.

	Minimum	Maximum	Mean (SD)	Median (IQR)	Cronbach’s Alpha
Overall WHOQOL-BREF	12.50	100.00	68.4 (14.49)	75 (62.5–75)	0.70
Physical health	27.78	97.22	65.56 (12.53)	66.67 (58.33–75)	0.82
Psychological health	4.17	91.67	64.31 (12.22)	66.67 (58.33–70.83)	0.79
Social relationships	15.00	90.00	66.56 (13.25)	70 (60–75)	0.74
Environment	21.88	81.25	53.76 (12.99)	53.13 (43.75–65.63)	0.76
Resilience score (CD-RISC)	35.00	100.00	66.38 (12.83)	66.5 (58–74)	0.93
Attack on personality	0.00	29.00	4.78 (6.37)	2 (0–6)	0.87
Attack on professional	0.00	28.00	5.74 (6.51)	3 (1–8)	0.86
Individual’s isolation from work	0.00	45.00	5.82 (7.61)	4 (1–7)	0.89
Direct attack	0.00	19.00	1.52 (3.38)	0 (0–2)	0.78
Total WPVB score	0.00	110.00	17.87 (21.12)	9 (5–23)	0.95

**Table 3 healthcare-13-01908-t003:** Correlation coefficients of WHOQOL-BREF subscales with CD-RISC and WPVB scales.

		Overall QoL	Physical Health	Psychological Health	Social Relationships	Environment
Resilience score (CD-RISC)	r	0.14	0.30	0.40	0.35	0.31
	*p*	0.189	0.004	<0.001	0.001	0.004
Attack on personality	rho	−0.08	0.03	−0.06	−0.02	−0.02
	*p*	0.436	0.789	0.585	0.887	0.886
Attack on professional	rho	−0.17	0.04	−0.08	−0.08	−0.17
	*p*	0.119	0.710	0.467	0.482	0.107
Individual’s isolation from work	rho	−0.01	−0.02	−0.06	−0.23	−0.33
	*p*	0.898	0.817	0.589	0.030	0.002
Direct attack	rho	−0.06	−0.04	−0.18	−0.16	−0.30
	*p*	0.604	0.679	0.092	0.137	0.005
Total WPVB score	rho	−0.14	−0.02	−0.12	−0.12	−0.28
	*p*	0.205	0.861	0.271	0.276	0.008

Note. Pearson’s correlation coefficients (r) and Spearman’s correlation coefficients (rho) are provided in the table.

**Table 4 healthcare-13-01908-t004:** Multiple linear regression analyses results with “overall WHOQOL-BREF”, “physical health”, and “psychological health” scores as dependent variables.

	Overall WHOQOL-BREF		Physical Health		Psychological Health	
	β (SE)+	*p*	β (SE)+	*p*	β (SE)+	*p*
Gender (Females vs. Males)	0.52 (4.22)	0.902	0.15 (3.52)	0.966	−2.22 (2.89)	0.443
Age	−0.18 (0.19)	0.348	−0.02 (0.16)	0.927	−0.16 (0.13)	0.231
Educational level	−1.16 (1.72)	0.505	1.6 (1.5)	0.290	1.7 (1.23)	0.171
Married (yes vs. no)	−3.75 (5.15)	0.469	0.94 (4.48)	0.835	−5.09 (3.67)	0.170
Have children (yes vs. no)	−1.81 (5.07)	0.722	−7.72 (4.39)	0.083	−2.85 (3.6)	0.430
Living (with others vs. alone)	13.1 (5.48)	0.019	12.41 (4.77)	0.011	13.51 (3.91)	0.001
Health problems (yes vs. no)	−6.18 (3.65)	0.095	−1.44 (3.18)	0.652	1.63 (2.6)	0.532
Resilience score (CD-RISC)	0.13 (0.12)	0.294	0.25 (0.11)	0.023	0.24 (0.09)	0.006
Total WPVB score	0.08 (0.08)	0.341	0.1 (0.07)	0.142	0.05 (0.06)	0.330

**Table 5 healthcare-13-01908-t005:** Multiple linear regression analyses. Results with “social relationships” and “environment” scores as dependent variables.

	Social Relationships		Environment	
	β (SE)+	*p*	β (SE)+	*p*
Gender (females vs. males)	−0.72 (3.64)	0.845	−2.24 (4.10)	0.588
Age	−0.09 (0.17)	0.609	0.00 (0.18)	0.994
Educational level	−1.20 (1.55)	0.441	2.48 (1.66)	0.139
Married (yes vs. no)	−0.78 (4.63)	0.866	8.43 (5.09)	0.102
Have children (yes vs. no)	−4.29 (4.54)	0.347	−8.98 (4.93)	0.073
Living (with others vs. alone)	9.43 (4.93)	0.059	−0.85 (5.48)	0.877
Health problems (yes vs. no)	4.21 (3.28)	0.204	4.18 (3.50)	0.236
Resilience score (CD-RISC)	0.32 (0.11)	0.005	0.27 (0.12)	0.024
Total WPVB score	0.05 (0.07)	0.494	−0.01 (0.08)	0.878

**Table 6 healthcare-13-01908-t006:** Mediation analysis results: resilience score as a mediator.

Predictor Variable	Outcome Variable	Indirect Effect (95% CI)	*p*-Value (Indirect)	Direct Effect (95% CI)	*p*-Value (Direct)	Mediation Significant?
Total WPVB Score	Environment	−0.053 (–0.153, 0.010)	0.10	−0.012 (–0.162, 0.138)	0.87	No
Direct Attack Score	Environment	−0.143 (–0.740, 0.172)	0.37	−0.359 (–0.740, 0.172)	0.18	No
Individual’s Isolation from Work Score	Environment	−0.124 (–0.371, 0.043)	0.14	−0.153 (–0.550, 0.245)	0.44	No
Individual’s Isolation from Work Score	Social Relationships	−0.148 (–0.464, 0.026)	0.09	0.057 (–0.317, 0.431)	0.76	No

## Data Availability

The data collected and analyzed during the current study are not publicly available due to privacy and ethical restrictions involving hospital staff participants. Data access is limited and governed by institutional approvals and informed consent agreements. Requests for data access may be considered upon reasonable request and subject to ethics approval.

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
