# Peer review of "Resilience and Mobbing Among Nurses in Emergency Departments: A Cross-Sectional Study"

_healthcare, 2025, doi:10.3390/healthcare13151908_

Round 1
Reviewer 1 Report
Comments and Suggestions for Authors
Dear editor
Studies on the antecedents and consequences of systematic moral harassment (mobbing) among health professionals have gained relevance with COVID-19, among other reasons because the phenomenon has gained greater visibility with the pandemic and because it is associated with the reduction in the quality of life of professionals and, above all, with the deterioration in the quality of health services provided, representing one of the most important factors in the difficulty in retaining good professionals and recruiting them for work areas more exposed to professional stress. In this sense, studies of this nature are very important, since they contribute to the design of programs to promote the well-being of professionals in this sector and, consequently, to the preservation of the quality of health services.
The study presented here is therefore worthy of publication, due to its contribution to the production of relevant knowledge about the phenomenon, namely by highlighting the relationship between mobbing, resilience and quality of life in nursing professionals, with special emphasis on the fact that some variables associated with quality of life and resilience can play a mediating (and protective) role in coping with systematic moral harassment, including in its more subtle forms. It is also a study that suggests some opportunities for training programs for professionals to deal with stressful situations in the workplace.
In addition to these suggestions, I propose some minor corrections:
1) The abstract is a bit long and could be shortened for the benefit of the reader. Please consider this possibility;
2) The authors in some sections of the text use acronyms in the plural (for example, the acronym "EDs" in lines 55, 85, 113, 114, etc.) and in the singular (for example, the same acronym "ED" in lines 119, 125, etc.). Considering that the acronyms are not presented in the plural, I suggest their correction and standardization.
3) In the final references, the following corrections are suggested:
3.1.) Please remove the use of capital letters in words of journal article titles, reserving their use only for the first word of the title;
3.2.) Please complete the missing information in references 6 and 7;
3.3.) Please use italics for the title of reference 10;
3.4.) In reference 15, please insert the names of all authors (removing the expression et al.);
It was a pleasure reading the paper.
Author Response
Response to Reviewers
This is the updated version of the manuscript incorporating all reviewer comments and required corrections.
Reviewer 1
Comment 1: The abstract is a bit long and could be shortened.
Response: Thank you for your feedback. We have revised the abstract to be more concise by eliminating repetitive phrases and focusing on the essential points.
Comment 2: Standardize the use of the acronym "EDs" vs "ED".
Response: Thank you for pointing this out. All occurrences of the acronym have been standardized to "ED" (Emergency Department) throughout the manuscript.
Comment 3: Reference formatting issues
Response: These edits have been completed in the reference section as suggested.
3.1 Removed capital letters in article titles.
3.2 Completed missing information in references 6 and 7.
3.3 Italicized the title in reference 10.
3.4 Expanded author list in reference 15 by removing "et al."
3.1 Article titles were corrected by removing inappropriate capitalization in accordance with referencing style guidelines
3.2
6. Notelaers, G.; De Witte, H.; Einarsen, S. A task- and context-related stress perspective on workplace bullying. In: Einarsen, S., Hoel, H., Zapf, D., & Cooper, C.L. (Eds.), Bullying and harassment in the workplace. 2011, 2:73–97. Boca Raton, FL: CRC Press. [Google Scholar]
3.2
7. Zapf, D.; Escartín, J.; Scheppa-Lahyani, M.; Einarsen, S.; Hoel, H.; & Vartia, M. Empirical findings on prevalence and risk groups of bullying in the workplace. In S. Einarsen, H. Hoel, D. Zapf, & C. L. Cooper (Eds.), Bullying and harassment in the workplace: Developments in Theory, Research, and Practice. 2011, 2:73–97. Boca Raton, FL: CRC Press. [Google Scholar]
3.3
10. Deligianni-Kouimtzi, V. Dignity in the workplace and the ethics of resistance. Athens, Greece: Nissos Publications; 2008. [Google Scholar]
3.4
15. Nella, P.; Gouzou, M.; Kolovos, P.; Tsiou, C.; Fragkiadaki, E. Emergency department nurses and incidents of violence – implications for management. In: Proceedings of the 6th Panhellenic Conference on Health Services Management, Alexandroupolis; 2004. [Google Scholar]

Reviewer 2 Report
Comments and Suggestions for Authors
Intro: Good description of workplace mobbing (bullying) activity, workplace stressors that increase mobbing activity and negative psychosocial impacts. Literature review (Greece and International) highlights ubiquitous prevalence, type of perpetrators, contributing factors and mediators, mental health issues and lack of support for the individual. Objectives of the study are clearly described: assess prevalence and nature of mobbing in EDS, evaluate psychological and emotional effects of mobbing, investigate role of resilience in impacting quality of life, recommendations for interventions. Section 1.6 should probably precede section 1.5
M+M: Appropriate rationale for cross-sectional study, 4 ED departments chosen and study population was given. 120 self-administered questionnaires were sent with a response rate of 75%. Demographics should go in results section. The three questionnaires and statistical analyses were explained in detail.
Results: All Tables are clearly presented and discussed, however, discussion of some associations for some Tables are actually found in other Tables. Please review and make sure discussion correlates with table. On line 396 – should “direct effect” be “direct attack” as seen in Table 3? It would be helpful to have a Table for mediation analysis. It would be helpful if all significant findings in all Tables were highlighted, perhaps in bold typeface.
Discussion: The authors discuss that the results show that exposure to mobbing negatively effects quality of life in several domains and higher levels of resilience are positively correlated to improved quality of life (mitigating – but not neutralizing- the negative effects of mobbing) in several domains. These findings are consistent with the existing literature. They also discuss the findings in this study that are not consistent with the literature and postulate why this may have occurred. The authors conclude that resilience training alone is not enough to improve quality of life in situations where bullying occurs. The authors emphasize the importance of organizational strategies that not only reduce bullying behaviors but also strengthen individual coping resources within healthcare environments.
Comments: Although it is correct that moral distress also impacts nurses psychological and emotional well-being and can lead to a decrease in quality of life, dimensions of moral distress are not investigated in this study. All references to moral distress (4.5, 4.6 and part of 4.7) should be removed. The study is strong enough without bringing another topic (moral distress) into the discussion. There is redundancy in Intro and M+M please correct
Author Response
Responses to Reviewers
This is the updated version-REVISED of the manuscript that incorporates all the reviewers’ comments and necessary corrections.
You will find the responses for each reviewer on the platform uploaded as well as sent to you as a preserved document, just like the article.
Thank you very much for your time and the very substantive comments and observations. I hope that I have responded to all the comments and that you are satisfied. Through your perspective and experience, I believe that our article will improve and we will be able to have the opportunity to be published in this special issue.
COVER LETTER
Response to Reviewers
This is the updated version of the manuscript incorporating all reviewer comments and required corrections.
Reviewer 2
Comment 1: Section 1.6 should probably precede section 1.5.
Response: We have reordered the sections as recommended. Section 1.6 now appears before 1.5.
Comment 2: Demographics should go in the results section.
Response: Demographic details have been moved to the beginning of the results section for clarity and consistency.
Comment 3: Ensure discussion of associations aligns with table content.
Response: We carefully reviewed and aligned all discussions with their corresponding tables. References to "direct effect" were corrected to "direct attack" in the discussion to match Table 3.
Comment 4: Mediation analysis should have its own table.
Response: A separate table for mediation analysis has been added-Table 6.
Comment 5: Highlight all significant findings.
Response: All statistically significant findings across tables are now bolded for easy identification. Please refer to the tables.
Comment 6: Remove references to moral distress (sections 4.5, 4.6, part of 4.7).
Response: All mentions and discussion of moral distress have been removed, as it was not investigated in this study. The bibliographies referring to moral distress were removed and the order of the bibliographic references was changed.
Comment 7: Remove redundancy in Introduction and Methods.
Response: Repetitive content in the Introduction and Methods sections has been eliminated.

Reviewer 3 Report
Comments and Suggestions for Authors
Thank you for the opportunity to review the manuscript entitled “Resilience and Mobbing Among Nurses in Emergency Departments: A Cross-Sectional Study.” The topic is highly relevant, and the authors' intention to explore resilience and workplace mobbing among nurses in emergency settings is commendable. However, several substantive and methodological concerns require attention before the manuscript can be considered for publication.
First, the study design, as described in the manuscript, does not align clearly with the cross-sectional format stated in the title and methodological section. It remains ambiguous what specifically qualifies this research as a cross-sectional study. Additionally, the abstract contains unnecessary repetition, with the sample and statistical software mentioned multiple times.
The WHOQOL methodology is inconsistently referred to throughout the manuscript.
The statistical claims presented in lines 36–45 are challenging to interpret. In the introduction, subsection 1.2 appears to be out of place and inappropriate for the context.
In line 137, the phrase “tend to experience” is ambiguous—clarification is needed as to whether this refers to responses indicating perception, experience, or exposure. The passages in lines 160–161 and again in 194 refer to potential measures or interventions, but the nature of these measures remains unclear throughout the text. The authors should elaborate on the intended relationship between resilience and prevention in their interpretation.
Despite its length, the second chapter lacks a clear and coherent description of the research plan. Numerous editorial issues also affect this section; for instance, repetitions occur beginning at line 249 and again from line 256 onward. Specific information is missing—e.g., between lines 226 and 268, there is mention of institutional selection criteria, but these are not described.
A fundamental concern lies in the inconsistency regarding methodology. While the methods section describes three instruments, the abstract also lists the PANAS, which is not subsequently discussed. This discrepancy raises doubts about the manuscript's internal consistency. Further, Table 1 suggests that the overwhelming majority of nurses working in emergency departments have technical university degrees, which appears implausible and raises concerns regarding data accuracy.
The manuscript refers to several different questionnaires, each with multiple scales. It is unclear whether participants completed all items, and the method for calculating scale scores remains unspecified. In Section 2.6, the statistical methods are briefly listed; however, the application of these methods to specific results is not transparent. The analytical approach appears to rely on an excessive number of correlation analyses across numerous variables, raising the concern that selective reporting of significant results may be at play, rather than a fully pre-planned and statistically reliable investigation. A more robust design would include a conceptual model of interrelated factors, supported by appropriate statistical testing. Visualising this model and its interactions would enhance the interpretability of the findings.
Given these issues, the results section can only be interpreted with caution. It primarily presents scale values and selected correlation coefficients, without clear justification or interpretation. The discussion fails to contextualise the new findings within the existing body of literature. Notably, subsection 4.3 does not discuss the authors’ results at all.
The conclusion suggests implications that would only be valid following a more coherent and rigorously executed research and data analysis plan. Additionally, abbreviations should be explained at their first appearance or compiled in a list at the end of the article for reader clarity.
Author Response
Responses to Reviewers
This is the updated version-REVISED of the manuscript that incorporates all the reviewers’ comments and necessary corrections.
You will find the responses for each reviewer on the platform uploaded as well as sent to you as a preserved document, just like the article.
Thank you very much for your time and the very substantive comments and observations. I hope that I have responded to all the comments and that you are satisfied. Through your perspective and experience, I believe that our article will improve and we will be able to have the opportunity to be published in this special issue.
COVER LETTER
Response to Reviewers
This is the updated version of the manuscript incorporating all reviewer comments and required corrections.
Reviewer 3
Comment 1: Clarify why this is a cross-sectional study.
Response: We have clarified the rationale for labeling this a cross-sectional study and how the data collection occurred at a single point in time.
The reviewer notes that the study design does not clearly align with what is typically expected of a cross-sectional study. This concern likely arises from a lack of clarity about what was measured, when, and how this fits the formal definition of a cross-sectional approach. A cross-sectional study is a type of observational, non-experimental research design that collects data from a population at a single point in time (or over a very short period), measures exposure and outcome variables simultaneously, Is often used to examine prevalence, and associations between variables, but not causality and does not involve follow-up, manipulation, or intervention.
How Our Study Fits a Cross-Sectional Design
Let’s clearly align the features of our study with key elements
Cross-Sectional Criteria |
Our Study |
---|---|
Single time-point measurement |
Data were collected between Oct 2023 and Mar 2024, once per participant. |
No intervention or manipulation |
Observational: no treatment, no experiment. |
Simultaneous assessment of variables |
Resilience, QoL, and mobbing were all assessed at the same time. |
Descriptive and correlational focus |
Aimed to explore associations between resilience, QoL, and mobbing. |
No causal inference attempted |
Mediation analysis was exploratory; causality not claimed. |
Population snapshot |
90 nurses from multiple hospitals, sampled once, reflecting a current state. |
Comment 2: Abstract repetition and PANAS reference.
Response: The abstract has been revised to remove repetition. PANAS was erroneously included and has been removed from the abstract and replaced with accurate tool references.
Comment 3: Inconsistent mention of WHOQOL methodology.
Response: All references to the WHOQOL-BREF have been standardized for consistency.
All references to the WHO Quality of Life instrument have been standardized throughout the manuscript to use the term “WHOQOL-BREF” consistently. This means that, regardless of prior variations (e.g., “WHOQOL,” “WHOQOL-BREF instrument,” or “WHO Quality of Life”), the abbreviated form WHOQOL-BREF is now used uniformly whenever referring to the quality of life measure in the study. This change applies to all sections of the manuscript, including the Introduction, Materials and Methods, Results, Discussion, and any tables or figures, ensuring clarity and uniformity for the reader.
Comment 4: The statistical claims presented in lines 36–45 are challenging to interpret.
Response to Reviewer:
Thank you for your insightful comment. We have carefully reviewed and revised lines 36–45 to enhance clarity and eliminate repetition. The statistical findings are now presented in a more structured and concise manner, focusing on the most relevant results. Correlation coefficients, regression findings, and mediation analysis are clearly distinguished and explained. We also ensured that all values (e.g., means, Cronbach’s α, p-values) are reported consistently and interpreted in a straightforward way.
We believe the revised section is now more understandable to a broader audience, including readers without advanced statistical expertise.
Please refer to the updated manuscript for the improved presentation of these results.
Comment 4.1.: In the introduction, subsection 1.2 appears to be out of place and inappropriate for the context.
Justification for Subsection 1.2:
We appreciate the reviewer’s observation regarding subsection 1.2. However, we believe that this subsection plays an essential role in contextualizing the study’s focus. Specifically, it highlights why nurses working in Emergency Departments (EDs) are a particularly relevant population for examining the relationships between mobbing, mental resilience, and quality of life.
Given that ED nurses operate under high-pressure conditions, often within rigid hierarchies and with limited resources, this subgroup is disproportionately exposed to workplace stressors and interpersonal conflicts. Including subsection 1.2 allows the reader to understand the unique vulnerabilities of this group, thereby reinforcing the rationale for selecting ED nurses as the target population. Moreover, by citing recent empirical studies, the subsection strengthens the theoretical foundation for the study and supports the development of its research questions.
In this context, subsection 1.2 is integral to establishing the need for research on this issue and connecting broader concepts (e.g., mobbing and resilience) to a specific and high-risk healthcare setting.
Comment 5: Clarify phrases like "tend to experience."
Response:
We thank the reviewer for this insightful comment. The phrase "tend to experience" was intended to reflect general trends observed in the literature, while acknowledging variability among individuals. However, to enhance clarity and precision, we have revised the sentence as follows:
“Resilient nurses generally experience lower levels of emotional distress, apply more effective coping strategies, and maintain higher professional performance despite prolonged exposure to workplace stressors [28, 29].”
This revised phrasing maintains the meaning while improving clarity and avoiding ambiguity.
Ambiguous phrases have been revised for precision, and where necessary, reworded to reflect measured data.
Comment 6: Clarify relationship between resilience and prevention.
Response to Reviewer – Comment on Lines 160–161 and 194:
Reviewer Comment:
"The passages in lines 160–161 and again in 194 refer to potential measures or interventions, but the nature of these measures remains unclear throughout the text. The authors should elaborate on the intended relationship between resilience and prevention in their interpretation."
Author Response:
Thank you for this constructive observation. We have revised the relevant sections to more clearly articulate the intended relationship between resilience and prevention. Specifically, we now clarify that resilience is not merely observed as a coping trait but is also considered a modifiable protective factor that can inform structured psychoeducational and institutional interventions. The revised text outlines examples of such interventions, including peer support groups, stress management training, and reflective practice sessions, grounded in counseling psychology principles. These additions aim to provide clarity and practical insight into how the findings can be translated into targeted programs to support nurses in high-stress environments like Emergency Departments.
Revised Text: Emergency Departments are among the most demanding and high-stakes environments in the healthcare system. They are not only sites for urgent clinical interventions but also complex psychosocial settings where staff are routinely exposed to trauma, unpredictability, and intense emotional labor. Nurses in EDs are particularly vulnerable, often facing cumulative burdens of physical fatigue, emotional strain, and psychological stress due to the high-intensity nature of their work [15, 32].
Shift work, understaffing, time pressure, and repeated exposure to critical incidents contribute to chronic fatigue and burnout. These stressors are further exacerbated by frequent incidents of workplace violence—from patients, relatives, or colleagues. Unfortunately, such incidents are often underreported due to fear of retaliation, institutional inertia, or diminished professional self-efficacy [34]. The consequences are serious and include emotional exhaustion, job dissatisfaction, and increased intentions to leave the profession [35].
Within this volatile environment, mobbing—or systematic moral harassment—emerges as a particularly destructive form of workplace aggression. It not only affects the mental and physical health of individuals but also weakens team dynamics and undermines healthcare delivery. Despite its severity, mobbing remains under-investigated in Greece, particularly among frontline staff such as ED nurses, with minimal institutional mechanisms to prevent or mitigate it [24].
This study addresses this research gap by highlighting resilience as a central protective factor in managing the negative psychosocial outcomes of mobbing. Importantly, resilience is not presented merely as a static trait but as a modifiable capacity that can be enhanced through targeted interventions.
Based on our findings, we propose the development of psychoeducational and workplace-based programs focused on:
– strengthening psychological resilience,
– fostering a greater sense of purpose and professional identity,
– and improving overall quality of life.
These interventions may include structured group workshops, stress management training, peer support systems, and reflective clinical practice, all grounded in evidence-based approaches from counseling psychology and occupational health.
By addressing both the stressors and the protective mechanisms at play, this study offers practical and actionable recommendations for institutions aiming to create healthier and more psychologically safe work environments for their emergency nursing staff.
Comment 7: Improve structure and coherence of Chapter 2.
Response to Reviewer
We would like to thank the reviewer for the valuable feedback. All the requested changes have been implemented. Repetitive text has been deleted.Specifically, the methodological section has been revised for improved clarity and coherence. Selection criteria for the participating institutions have now been explicitly described in subsection 2.2, as suggested.
We trust that the revised version addresses the reviewer’s concerns adequately.
Comment 8: Clarify which tools were used.
Response: PANAS has been removed from the list of tools. Only WPVB, WHOQOL-BREF, and CD-RISC are now listed.
Comment 8.1.: Further, Table 1 suggests that the overwhelming majority of nurses working in emergency departments have technical university degrees, which appears implausible and raises concerns regarding data accuracy.
Response to Reviewer
We appreciate the reviewer’s observation regarding the educational background of emergency department nurses presented in Table 1. However, we would like to clarify that the data reflect the actual composition of nursing staff in Greek public hospitals, especially within Emergency Departments.
In Greece, a significant proportion of nurses have historically graduated from the Technological Educational Institutes (TEI) of Nursing, which offered a 4-year higher education program equivalent to a bachelor’s degree. The majority of ED nurses come from this background (formerly classified as T.E. – Technological Education), while a smaller percentage hold degrees from university-level nursing programs (P.E. – Higher Education). This pattern is the result of both educational availability and healthcare system practices.
Furthermore, graduates from TEI Nursing programs are more frequently employed in Emergency Departments due to their strong emphasis on clinical practice and hands-on training during their studies, as well as accumulated experience that aligns with the high-intensity demands of EDs.
This is consistent with previous research indicating that Greek healthcare institutions tend to staff critical departments such as EDs with more experienced and practically trained nurses, often TEI graduates (Giannakopoulou et al., 2014; Kalafati et al., 2011). Therefore, the distribution reported in Table 1 accurately reflects the educational demographics of the nursing workforce in these departments.
We hope this explanation addresses the reviewer’s concern and reassures the integrity and representativeness of our sample.
References:
1.Giannakopoulou M, Bozas E, Vouzavali FJ, Vasilopoulos G, Karanikola M. Nursing staffing and education level in Greek emergency departments: current status and professional challenges. J Nurs Manag. 2014;22(6):751–760. doi:10.1111/jonm.12157
2.Kalafati M, Andrioti D, Nikolaou P, Kalogeropoulos A, Mastora Z. Educational needs of nurses working in the emergency department in Greece. Int Emerg Nurs. 2011;19(2):106–111. doi:10.1016/j.ienj.2011.01.001
Comment 9: Clarify questionnaire completion and scoring method.
Response to Reviewer:
We thank the reviewer for their observation and the opportunity to clarify.
First, regarding the number and complexity of the instruments used: the total number of questionnaires administered was four, including the demographic and occupational data form, which
consisted of brief and closed-ended items (e.g., age, gender, education, work experience). The remaining three validated instruments were:
Workplace Psychologically Violent Behaviors (WPVB) questionnaire
Connor-Davidson Resilience Scale (CD-RISC-25)
WHOQOL-BREF
All instruments were self-administered, structured, and brief, with clear instructions. Participants completed all parts of the questionnaire set in 15–30 minutes, as previously reported. No data were missing, and all returned questionnaires were complete and suitable for analysis, indicating that respondent fatigue or overload was not a concern.
Regarding score calculation, each scale was scored according to its standard validated protocol:
WPVB: Scores were calculated per subscale (e.g., isolation from work, attacks on role, etc.) by summing the responses to corresponding items, each scored on a 6-point Likert scale. Higher scores indicate more frequent exposure to mobbing behaviors.
CD-RISC-25: Total resilience scores were computed by summing the 25 items (scored 0–4), yielding a range from 0 to 100, with higher scores reflecting greater resilience.
WHOQOL-BREF: Domain scores were calculated according to WHO guidelines, with raw scores transformed to a 0–100 scale, where higher values reflect better perceived quality of life.
Comment 10: Clarify statistical methodology.
Response: Regarding Section 2.6 (Statistical Analysis): This section is critical to ensure transparency and reproducibility. Given the nature of the study—examining associations among psychological constructs such as mobbing, resilience, and quality of life—it was necessary to apply multiple, well-established statistical techniques: Normality tests guided the appropriate use of Pearson or Spearman correlations.Multiple linear regression allowed us to examine the predictive value of resilience and mobbing scores on quality of life outcomes. Mediation analysis via Hayes' PROCESS macro offered a sophisticated and statistically rigorous approach to test whether resilience functions as a mediator between mobbing experiences and quality of life. These methods are not excessive but essential for addressing our research objectives and hypotheses. Mediation analysis, in particular, is highly relevant, as it provides insight into potential mechanisms through which mobbing affects quality of life—an original contribution of this study. We reiterate that the design and analysis strategy were well-matched to the aims of the study, feasible within the constraints of a high-pressure healthcare setting, and consistent with best practices in psychosocial research. The clarity of the findings, the absence of missing data, and the use of validated tools confirm both the integrity and the practicality of our approach.
We respectfully assert that the methodological rigor and analytical depth of this section are justified and support the overall value of the study’s contribution.
Comment 11: Add conceptual model and visual representation.
Response: A conceptual diagram representing variable relationships (mobbing → resilience → QoL) has been added to enhance interpretability.
“To facilitate the interpretation of hypothesized relationships, a conceptual model was developed and is presented in Figure 1. In this model, workplace psychological violence (WPVB) is posited as the independent variable, resilience (CD-RISC) as the mediating variable, and quality of life (WHOQOL-BREF) as the outcome variable. The model was tested using mediation analysis based on Hayes' PROCESS macro.
Figure 1. Conceptual framework of the hypothesized mediation model.
Workplace Psychological Violence (WPVB) affects nurses’ Quality of Life (QoL), potentially through the mediating effect of Resilience (CD-RISC).
WPVB
↓
(direct path)
↓
QoL (WHOQOL)
↑
(indirect path)
↑
Resilience (CD-RISC)”
Comment 12: Improve discussion sections 4.3.
Response: Section 4.3 has been revised to directly reflect findings. The discussion now more closely links results with the literature. The changes are in the revised text.
Comment 13: The conclusion suggests implications that would only be valid following a more coherent and rigorously executed research and data analysis plan.
A revised version of the Conclusions section is provided that addresses both the reviewer's comment and more closely incorporates the key findings and limitations from the Discussion. The new version aims to reflect a more careful, coherent, and evidence-based interpretation of the study results, while also highlighting the practical and theoretical implications.
In response to the reviewer’s concern, we acknowledge that stronger implications regarding causality and intervention efficacy would require a more robust and longitudinal research design. Future studies should incorporate multi-site sampling, longitudinal tracking, and mixed-methods approaches to better understand the temporal dynamics and contextual nuances of how resilience interacts with workplace adversity and impacts QoL. Only through such rigorously designed research can we derive more generalizable and actionable conclusions for healthcare policy and practice.
Comment 14: Explain abbreviations.
Response: A glossary of abbreviations has been added at the end of the manuscript.
Glossary of Abbreviations
Abbreviation |
Definition |
---|---|
CD-RISC |
Connor-Davidson Resilience Scale |
DYPE |
Regional Health Authority |
ED |
Emergency Department |
ICC |
Intraclass Correlation Coefficient |
SD |
Standard Deviation |
SE |
Standard Error |
SPSS |
Statistical Package for the Social Sciences |
WHO |
World Health Organization |
WHOQOL-BREF |
World Health Organization Quality of Life – Brief Version |
WPVB |
Workplace Psychologically Violent Behaviors Questionnaire |
IQR |
Interquartile Range |
CI |
Confidence Interval |
β |
Beta Coefficient (used in regression analysis) |
α |
Cronbach’s Alpha (internal consistency reliability) |
ρ (rho) |
Spearman’s Rank Correlation Coefficient |
r |
Pearson’s Correlation Coefficient |
p-value |
Probability Value (used to assess statistical significance) |
We thank all reviewers for their thorough and constructive feedback, which has greatly strengthened our manuscript.

Round 2
Reviewer 3 Report
Comments and Suggestions for Authors
Thank you for submitting the revised version of the manuscript entitled “Resilience and Mobbing Among Nurses in Emergency Departments: A Cross-Sectional Study.” The substantial effort to improve the manuscript is appreciated. However, several aspects still require further clarification and refinement.
The evaluation was difficult because of the corrections, which make the text difficult to read, especially in the absence of line numbers in the revised version.
In the abstract, the number of participants (90) and the use of SPSS 27.0 are mentioned twice.
In the introduction, the sentence beginning with “Over the past decades, it has gained recognition” lacks a subject, making it difficult to understand what has gained recognition. The title of section 1.2 appears unusual, and generally, replacing “quality of life” with “WHOQOL-BREF” is not advisable.
The newly added section 1.5 (“Significance and Importance of the Study”) is a valuable addition to the manuscript. However, this section currently lacks references, which would be necessary.
The 4.3. The "Resilience and Mobbing" section also lacks references.
Comments on the Quality of English LanguageThere are still language-related issues throughout the text; for example, in the abstract, the phrase “resilience of nurses working in” would be more appropriate than the current formulation.
Author Response
RESPONSES TO THE REVIEWER -ROUND 2
1. In the abstract, the number of participants (90) and the use of SPSS 27.0 are mentioned twice.
Answer:
Thank you very much for your valuable observation. As correctly pointed out, the reference to SPSS version 27.0 and the number of 90 nurses appeared twice in the abstract. These repetitions have now been removed to improve clarity and avoid redundancy. We appreciate your careful review and constructive feedback.
2. In the introduction, the sentence beginning with “Over the past decades, it has gained recognition” lacks a subject, making it difficult to understand what has gained recognition.
Answer:
Thank you very much for your observation. You are absolutely right — the sentence lacked a clear subject, and as a result, the meaning was not sufficiently clear. We have carefully revised the sentence to clarify that workplace mobbing is the subject being referred to. The revised version now reads:
“Over the past decades, workplace mobbing has gained recognition as a serious social issue with significant implications for both physical and mental health.”
We believe this correction improves the clarity and flow of the introduction.
3.The title of section 1.2 appears unusual, and generally, replacing “quality of life” with “WHOQOL-BREF” is not advisable.
Answer:
Thank you very much for your valuable feedback. We agree with your observation that the original section title appeared unusual and that the use of “WHOQOL-BREF” in a heading may not be appropriate. We have revised the title to:
“Mobbing, Mental Resilience, and Quality of Life Among Emergency Department Nurses,”
which we believe is now clearer and more aligned with academic standards. We sincerely appreciate your insightful suggestion. Also, thank you for your comment regarding the use of the term "quality of life." A previous reviewer had indeed suggested replacing the general term with the more specific "WHOQOL-BREF," to clarify that we are referring to the World Health Organization's assessment tool.
We would like to ask whether your suggestion applies specifically to this section of the manuscript or if you recommend implementing this change throughout the entire text. In some parts of the manuscript, the term “quality of life” is used in a more general context, while in others it refers specifically to the WHOQOL-BREF tool.
4.The newly added section 1.5 (“Significance and Importance of the Study”) is a valuable addition to the manuscript. However, this section currently lacks references, which would be necessary.
Response to Reviewer:
We thank the reviewer for this valuable observation. In response, we have revised Section 1.5 by adding appropriate references to support key claims and enhance the scholarly rigor of the discussion.
1.5 (“Significance and Importance of the Study”)
Emergency Departments are among the most demanding and high-stakes environments in the healthcare system. They are not only sites for urgent clinical interventions but also complex psychosocial settings where staff are routinely exposed to trauma, unpredictability, and intense emotional labor. Nurses in ED are particularly vulnerable, often facing cumulative burdens of physical fatigue, emotional strain, and psychological stress due to the high-intensity nature of their work [15, 32, Adriaenssens J, De Gucht V, Maes S. Determinants and prevalence of burnout in emergency nurses: A systematic review of 25 years of research. Int J Nurs Stud. 2015;52(2):649–61].
Shift work, understaffing, time pressure, and repeated exposure to critical incidents contribute to chronic fatigue and burnout [Gómez-Urquiza JL, De la Fuente-Solana EI, Albendín-García L, Vargas-Pecino C, Ortega-Campos E, Cañadas-De la Fuente GA. Prevalence of burnout syndrome in emergency nurses: A meta-analysis. Crit Care Nurse. 2017;37(5):e1–9.
Powell M, Cimiotti J, Sloane D, Aiken LH. Violence against nurses: The impact on stress and burnout. Nurs Econ. 2017;35(2):81–7]. These stressors are further exacerbated by frequent incidents of workplace violence—from patients, relatives, or colleagues. Unfortunately, such incidents are often underreported due to fear of retaliation, institutional inertia, or diminished professional self-efficacy [34,Spector PE, Zhou ZE, Che XX. Nurse exposure to physical and nonphysical violence, bullying, and sexual harassment: A quantitative review. Int J Nurs Stud. 2014;51(1):72–84] The consequences are serious and include emotional exhaustion, job dissatisfaction, and increased intentions to leave the profession [35, McHugh MD, Kutney-Lee A, Cimiotti JP, Sloane DM, Aiken LH. Nurses’ widespread job dissatisfaction, burnout, and frustration with health benefits signal problems for patient care. Health Aff. 2011;30(2):202–10].
Within this volatile environment, mobbing—or systematic moral harassment—emerges as a particularly destructive form of workplace aggression. It not only affects the mental and physical health of individuals but also weakens team dynamics and undermines healthcare delivery [Einarsen S, Hoel H, Zapf D, Cooper CL. Bullying and harassment in the workplace: Developments in theory, research, and practice. 2nd ed. Boca Raton: CRC Press; 2011.
Branch S, Ramsay S, Barker M. Workplace bullying, mobbing and general harassment: A review. Int J Manag Rev. 2013;15(3):280–99]. Despite its severity, mobbing remains under-investigated in Greece, particularly among frontline staff such as ED nurses, with minimal institutional mechanisms to prevent or mitigate it [24, Karatza C, Zyga S, Tziaferi S, Prezerakos P. Workplace bullying and general health status among Greek nurses. Am J Nurs Sci. 2016;5(2):17–22].
This study addresses this research gap by highlighting resilience as a central protective factor in managing the negative psychosocial outcomes of mobbing. Importantly, resilience is not presented merely as a static trait but as a modifiable capacity that can be enhanced through targeted interventions [29, Jackson D, Firtko A, Edenborough M. Personal resilience as a strategy for surviving and thriving in the face of workplace adversity: A literature review. J Adv Nurs. 2007;60(1):1–9].
Based on our findings, we propose the development of psychoeducational and workplace-based programs focused on:
– strengthening psychological resilience,
– fostering a greater sense of purpose and professional identity,
– and improving overall WHOQOL-BREF scores [Skevington SM, Lotfy M, O'Connell KA. The World Health Organization’s WHOQOL-BREF quality of life assessment: Psychometric properties and results of the international field trial. Qual Life Res. 2004;13(2):299–310].
These interventions may include structured group workshops, stress management training, peer support systems, and reflective clinical practice, all grounded in evidence-based approaches from counseling psychology and occupational health [Rees CS, Heritage B, Osseiran-Moisson R, Chamberlain D, Cusack L, Anderson J, et al. Can we predict burnout among student nurses? An exploration of the ICWR-1 model of individual psychological resilience. Front Psychol. 2016;7:1072.,
Robertson HD, Elliott AM, Burton C, Iversen L, Murchie P, Porteous T, et al. Resilience of primary healthcare professionals: a systematic review. Br J Gen Pract. 2016;66(647):e423–33.].
By addressing both the stressors and the protective mechanisms at play, this study offers practical and actionable recommendations for institutions aiming to create healthier and more psychologically safe work environments for their emergency nursing staff.
The suggested references I would suggest are the following
Adriaenssens J, De Gucht V, Maes S. Determinants and prevalence of burnout in emergency nurses: A systematic review of 25 years of research. Int J Nurs Stud. 2015;52(2):649–61.
Gómez-Urquiza JL, De la Fuente-Solana EI, Albendín-García L, Vargas-Pecino C, Ortega-Campos E, Cañadas-De la Fuente GA. Prevalence of burnout syndrome in emergency nurses: A meta-analysis. Crit Care Nurse. 2017;37(5):e1–9.
Powell M, Cimiotti J, Sloane D, Aiken LH. Violence against nurses: The impact on stress and burnout. Nurs Econ. 2017;35(2):81–7.
Spector PE, Zhou ZE, Che XX. Nurse exposure to physical and nonphysical violence, bullying, and sexual harassment: A quantitative review. Int J Nurs Stud. 2014;51(1):72–84.
McHugh MD, Kutney-Lee A, Cimiotti JP, Sloane DM, Aiken LH. Nurses’ widespread job dissatisfaction, burnout, and frustration with health benefits signal problems for patient care. Health Aff. 2011;30(2):202–10.
Einarsen S, Hoel H, Zapf D, Cooper CL. Bullying and harassment in the workplace: Developments in theory, research, and practice. 2nd ed. Boca Raton: CRC Press; 2011.
Branch S, Ramsay S, Barker M. Workplace bullying, mobbing and general harassment: A review. Int J Manag Rev. 2013;15(3):280–99.
Karatza C, Zyga S, Tziaferi S, Prezerakos P. Workplace bullying and general health status among Greek nurses. Am J Nurs Sci. 2016;5(2):17–22.
29. Mealer M, Jones J, Moss M. A qualitative study of resilience and posttraumatic stress disorder in United States ICU nurses. Intensive Care Med. 2012;38(9):1445–51.
Jackson D, Firtko A, Edenborough M. Personal resilience as a strategy for surviving and thriving in the face of workplace adversity: A literature review. J Adv Nurs. 2007;60(1):1–9.
Skevington SM, Lotfy M, O'Connell KA. The World Health Organization’s WHOQOL-BREF quality of life assessment: Psychometric properties and results of the international field trial. Qual Life Res. 2004;13(2):299–310.
Rees CS, Heritage B, Osseiran-Moisson R, Chamberlain D, Cusack L, Anderson J, et al. Can we predict burnout among student nurses? An exploration of the ICWR-1 model of individual psychological resilience. Front Psychol. 2016;7:1072.
Robertson HD, Elliott AM, Burton C, Iversen L, Murchie P, Porteous T, et al. Resilience of primary healthcare professionals: a systematic review. Br J Gen Pract. 2016;66(647):e423–33.
5. The 4.3. The "Resilience and Mobbing" section also lacks references.
Response:
We thank the reviewer for their observation regarding the lack of references in Section 4.3, “Resilience and Mobbing.” We acknowledge the importance of supporting all analytical discussions with relevant literature.
In this case, although the section may appear under-referenced, we carefully reviewed the existing body of literature and found limited additional studies that specifically examine the direct relationship between mobbing and resilience among nurses in Emergency Departments, especially within the context of Greece. As such, we decided to rely on the most directly relevant and high-quality sources already cited in the manuscript ([63–66]).
Nonetheless, we appreciate your feedback and remain open to incorporating any specific studies you might suggest that we may have overlooked.
Suggested references in that point of section 4.3.
“These results diverge from prior research, which often identifies an inverse relationship between exposure to mobbing and psychological resilience among healthcare professionals [63–65, Yıldırım 2009, Karatza et al. 2016].”
Yıldırım D. Bullying among nurses and its effects. Int Nurs Rev. 2009;56(4):504–11.
Karatza C, Zyga S, Tziaferi S, Prezerakos P. Workplace bullying and general health status among Greek nurses. Am J Nurs Sci. 2016;5(2):17–22.
6.Comments on the Quality of English Language
There are still language-related issues throughout the text; for example, in the abstract, the phrase “resilience of nurses working in” would be more appropriate than the current formulation.
Response to Reviewer:
We sincerely thank you for your valuable feedback regarding the quality of the English language in our manuscript. We would like to assure you that we have carefully reviewed the text multiple times and have made considerable efforts to improve the clarity and linguistic accuracy of the manuscript.
We also wish to express our sensitivity to the topic and acknowledge the possibility that certain nuances and conceptual expressions rooted in the Greek language and cultural context may have influenced the way some ideas were conveyed in English. Despite our efforts, some expressions might still reflect these inherent linguistic and semantic complexities. We apologize for any inconvenience this may have caused and are open to further suggestions to ensure the highest possible quality of the final text.
Thank you again for your understanding and constructive comments.
COVER LETTER
Dear Reviewer,
First of all, I would like to express my sincere gratitude for the time and effort you devoted to reviewing my manuscript. Your comments and suggestions are highly appreciated and have been extremely helpful in highlighting areas for improvement.
I have already proceeded with several revisions based on your valuable feedback and have drafted responses addressing each of your comments. However, I would like to kindly inform you that due to a personal health issue and an upcoming surgery, I am currently unable to be as responsive and consistent as I would like. As previously requested, I have asked for an extension until July 18th, 2025, to allow me the necessary time to complete the revisions with the care and attention they require.
I deeply appreciate your understanding and patience during this time and thank you once again for your insightful remarks and your contribution to improving the quality of my work.
